# Activated Carbon Derived from Cucumber Peel for Use as a Supercapacitor Electrode Material

**DOI:** 10.3390/nano14080686

**Published:** 2024-04-16

**Authors:** Meruyert Nazhipkyzy, Gulim Kurmanbayeva, Aigerim Seitkazinova, Esin Apaydın Varol, Wanlu Li, Balaussa Dinistanova, Almagul Issanbekova, Togzhan Mashan

**Affiliations:** 1Department of Chemical Physics and Material Science, Al-Farabi Kazakh National University, 71 Al-Farabi Ave., Almaty 050038, Kazakhstanaikolove00@mail.ru (A.S.); 2Institute of Combustion Problems, Bogenbai Batyr Street 172, Almaty 050012, Kazakhstan2012_kurator@mail.ru (A.I.); 3Department of Materials Science, Nanotechnology and Engineering Physics, Satbayev University, Satpaev St. 22, Almaty 050000, Kazakhstan; 4Department of Chemical Engineering, Eskisehir Technical University, Eskişehir 26555, Turkey; eapaydin@eskisehir.edu.tr; 5Department of Chemistry and Biochemistry, Montclair State University, 1 Normal Ave., Montclair, NJ 07043, USA; 6UNESCO Chair in Sustainable Development, Al-Farabi Kazakh National University, 71 Al-Farabi Ave., Almaty 050038, Kazakhstan; 7Department of Chemistry, L.N. Gumilyov Eurasian National University, Kazhymukan Str. 11, Astana 010000, Kazakhstan; togzhan-mashan@mail.ru

**Keywords:** biowaste, capacitive performance, porous carbon, pore accessibility, cucumber peels

## Abstract

Biowaste conversion into activated carbon is a sustainable and inexpensive approach that relieves the pressure on its disposal. Here, we prepared micro-mesoporous activated carbons (ACs) from cucumber peels through carbonization at 600 °C followed by thermal activation at different temperatures. The ACs were tested as supercapacitors for the first time. The carbon activated at 800 °C (ACP-800) showed a high specific capacitance value of 300 F/g at a scan rate of 5 mV/s in the cyclic voltammetry and 331 F/g at the current density of 0.1 A/g in the galvanostatic charge–discharge analysis. At the current density of 1 A/g, the specific discharge capacitance was 286 F/g and retained 100% capacity after 2000 cycles. Their properties were analyzed by scanning electron microscopy, energy-dispersive X-ray analysis, porosity, thermal analysis, and Fourier-transform infrared spectroscopy. The specific surface area of this sample was calculated to be 2333 m^2^ g^−1^ using the Brunauer–Emmett–Teller method. The excellent performance of ACP-800 is mainly attributed to its hierarchical porosity, as the mesopores provide connectivity between the micropores and improve the capacitive performance. These electrochemical properties enable this carbon material prepared from cucumber peels to be a potential source for supercapacitor materials.

## Highlights

Cucumber-peel-derived carbons were tested as a supercapacitor for the first time.The carbon showed a high specific capacitance of 286 F/g at 1 A/g.The carbon activated at 800 °C displayed a high specific surface area (2333 m^2^/g).

## 1. Introduction

One of the most critical environmental protection issues is to develop a sustainable system to provide electrical energy. Conventional energy sources such as fossil fuels (coal, oil, and gas) are not only consumed at a rapid rate but are also accompanied by the destruction of ecosystems, loss of wildlife, and environmental pollution. Developing renewable energy can be accelerated by an efficient energy storage system. Energy storage and delivery technologies using supercapacitors can store and transmit energy at high rates and charge high current densities quickly [1]. Besides, supercapacitors have a virtually unlimited charge cycle at high power density and perform better than batteries at extreme temperatures [1]. Research on electrochemical energy storage devices such as supercapacitors and battery storage has been widely explored. The design of hybrid supercapacitors that apply capacitive materials as the negative electrode and battery-type materials as the positive electrode is emerging and attractive since the device can combine the fast charging/discharging and long lifespan of capacitive materials with the high capacity of battery-type materials [2,3]. It is no exaggeration to say that the effective implementation of any renewable energy source, hybrid or electric vehicles, and smart grids depends heavily on the availability of a suitable energy storage system.

Using renewable materials as supercapacitor electrodes is of great value since the materials have a low price and do not harm the environment [4]. Porous carbons from biomass have been intensively studied as materials for supercapacitor electrodes and exhibit excellent electrochemical properties. Their high specific surface area and large pore size are crucial for improving the electrochemical performance of carbon-based supercapacitors. Porous carbon materials for supercapacitor electrodes are mainly obtained by pyrolysis at a temperature of 600–1000 °C in an inert atmosphere. Methods such as physical activation (using water vapor, CO_2_, or air) and chemical activation (NaOH, KOH, ZnCl_2_, HNO_3_, and H_2_SO_4_) are widely used to increase the porosity and surface area of carbon materials during pyrolysis [5].

Various peels of biowaste have been used as precursors to produce activated porous carbon, such as mangosteen, cassava, watermelon, onion, orange, lemon, pomelo, garlic, banana [6,7,8,9,10,11,12,13,14,15,16,17,18], coconut [19,20,21,22,23,24], apricot [25,26,27], palm kernel [28,29,30,31,32,33,34], etc. Compared to conventional carbon sources, these biomass materials are copious, natural, and renewable, making them sustainable and inexpensive carbon sources. In the work of the Fasakin group [16], the electrode material was made from banana peels activated by potassium hydroxide at a temperature of 900 °C. The specific capacity was reported to be 165 F/g at a current density of 0.5 A/g in a two-electrode cell configuration. The sample obtained under the above conditions demonstrated such material properties as a hierarchical porous nanostructure containing micropores and mesopores with the largest specific surface area (1362 m^2^/g). Another study of banana-peel-derived carbon [17] involved carbonization in the presence of melamine (N-dopant), sodium thiosulfate (S-dopant), and KOH activation to design novel N, S-co-doped hierarchically porous carbonaceous materials. The synthesized sample exhibited an interconnected porosity endowed with a high specific surface area (2452 m^2^/g) and demonstrated a specific capacitance of 220 F/g at 0.5 A/g. Even though the carbons were obtained from the same biomass source, the nitrogen and sulfur dopants played a collaborative role in the chemical activation by forming a high surface area and optimum pores. In the work of Ashraf et al. [24], steam-activated carbon obtained from coconut was tested in a two-electrode cell. The group tested three acid-washing procedures, and the best-performing carbon was the one washed with hydrofluoric acid, showing a specific capacity of 162 F/g at a current density of 1 A/g. Treatment with HF yielded an activated carbon sample with high purity and a surface area of 1864 m^2^/g that was 63% mesoporous, which is suitable for facilitating the rapid diffusion and transport of electrolyte ions. Shu et al. [26] obtained hierarchically porous activated carbons from apricot shells through carbonization followed by steam activation. As a result, activated carbon with a surface area of 1240 m^2^/g that was 26% mesoporous showed a specific capacity of 38 F/g at 0.05 A/g in a two-electrode cell. It also demonstrated stability over 6000 cycles at 5 A/g with 99% retention. Steam activation of different biomass sources leads to dramatic changes in porous structure and capacitive value. Misnon et al. [28] carbonized the shells of oil palms and subsequently activated them by chemical methods (KOH, 6 M) and physical methods (steam activation). As a result, the chemically activated carbon exhibited a specific capacity of 210 F/g at 0.5 A/g, while that of the physically activated carbon was 50% lower (123 F/g) than that of the chemically activated carbon. The authors stated that the chemically activated carbon had micropores and mesopores, which facilitated ion transfer from mesopores to micropores and improved its electrochemical characteristics, especially the charge–discharge rate. The physically activated carbon possessed only micropores, which limited the incorporation/extraction of ions on the carbon surface. In the work of Gou et al. [13], nitrogen-doped porous carbon (PN-OPC) was synthesized from orange peels. The micro-mesoporous network structure and nitrogen contents (7.8%) resulted in the rapid penetration of electrolyte ions and electron transmission, giving the PN-OPC electrode material a high specific capacity of 255 F/g at a current density of 0.5 A/g. The biomass source, activation methods (chemical or physical), addition of dopants, and acid washing can all play a role in the degree of carbon’s hierarchy of porous structure, thus affecting their capacitive performance. Gopalakrishnan [35] synthesized graphene-like carbon nanosheets by carrying out hydrothermal pre-carbonization followed by the pyrolysis of cucumber. In the two-electrode cell setup, a specific capacity of 143 F/g was reported at a current density of 0.2 A/g in 6 M KOH. Whereas a specific capacity decreased to 58 F/g at a current density of 0.025 A/g. The high-capacity retention was 97% after 1000 cycles. The formation of a creasy graphene-like structure in the samples was one of the reasons for their higher specific capacity. The micropores and mesopores in the carbon nanosheets also provided a fast and convenient path for ion transport. Park’s group recently prepared N-doped microporous carbons from cucumber peels, proving them to be efficient for CO_2_ capture [36].

The abundance of specific agricultural biomass depends on geographic features and farming choices. Like other peels of vegetables and fruits, cucumber peels are rich in cellulose. However, no study has explored the potential of cucumber peels as a carbon source to be applied to energy storage. In the present work, for the first time, micro-mesoporous carbons based on cucumber peels were synthesized and tested as electrode materials for a supercapacitor. By characterizing the surface chemistry and porosity of the carbons, the reasons for their excellent electrochemical performance were explored. The main objectives of the work were to demonstrate a facile route to obtain activated nanoporous carbons from cucumber peels and their excellent electrochemical performance.

## 2. Material and Methods

### 2.1. Activated Carbon Preparation

The raw material cucumber peel used in this study was obtained from Almaty, Kazakhstan. Cucumber peels were dried at atmospheric conditions and then ground to obtain an average particle size. Activated carbon samples were prepared from the washed and dried (at 80 °C) cucumber peel by carbonization followed by the activation process, as expressed in Figure 1. The first thermal treatment, namely carbonization, was carried out at 600 °C for 2 h. A tubular reactor was used, and argon gas (99.993% Ikhsan Technogas Ltd., Almaty, Kazakhstan, 100 cm^3^/min) was preferred to maintain the inert atmosphere. The carbonaceous sample obtained after this process was designated as carbonated cucumber peel (CCP).

For the activation step, the CCP sample was thoroughly mixed with potassium hydroxide (KOH, ≥85%, Sigma Aldrich, St. Louis, MI, USA) in a mass ratio of 1:2. To complete the chemical activation, the CCP-KOH sample was placed in a tubular furnace and the second thermal treatment at 700 °C or 800 °C for 2 h was accomplished under argon gas flow. The activated carbon samples were washed with 1 M HCl and then with distilled water until reaching a stable pH. These samples were dried in an oven overnight. The two activated carbons were designated as activated cucumber peel 700 (ACP-700) and activated cucumber peel 800 (ACP-800). The carbon yield (%) was calculated by dividing the mass of the activated carbon (g) by the mass of dried cucumber peels (g).

### 2.2. Characterization

The surface morphology of CCP, ACP-700, and ACP-800 was measured by scanning electron microscopy (SEM, JEOL, model JSM-6490LA, FEI, Hillsboro, OR, USA). In addition, energy-dispersive X-ray analysis (EDAX, JSM-6490LA, FEI, USA) was applied to determine the elemental composition of the samples.

An adsorption analyzer (SORBTOMETR-M, Catakon, Novosibirsk, Russia) was used to obtain nitrogen adsorption isotherms at −196 °C. From the adsorption data, the surface area was calculated using the Brunauer–Emmett–Teller (BET) model. The total pore volume (V_t_) was calculated when the relative pressure was at 0.99. The 2D-NLDFT method was used to calculate the micropore volumes and pore size distributions by assuming the heterogeneity of the pore wall [37].

The thermal decomposition behavior of the samples was determined using thermogravimetric analysis (Setaram Labsys Evo TGA, Caluire, France). Initially, 10 ± 0.5 mg of sample was placed in a 100 µL alumina crucible and heated from 25 to 1000 °C at a heating rate of 20 °C/min under an N_2_ atmosphere (20 cm^3^/min).

For the Fourier-transform infrared spectroscopy (FT-IR, Perkin Elmer, Singapore) analysis, the KBr pellets were made using the mass ratio of 1% sample to 99% KBr. Then, the spectra were taken in the wavenumber range of 4000–400 cm^−1^.

The Solver spectrum instrument (NT-MDT, Moscow, Russia) was used to measure Raman spectra with the help of a 473 nm laser. The laser beam was directed at the sample using a 100 × 0.75 NA Mitutoyo lens, providing a laser spot < 2 µm in diameter.

### 2.3. Electrochemical Measurements

#### 2.3.1. Electrode Preparation

For the preparation of activated carbon electrodes, 75 wt.% activated carbon, 15 wt.% binders (polyvinylidene fluoride (PVDF), EQ-Lib-PVDF, MTI Corporation, Richmond, CA, USA), 10 wt.% carbon black (EQ-Lib-SuperC45, MTI Corporation), and 1–2 mL of 1-methyl-2-pyrrolidone (≥99.0%, Sigma Aldrich, St. Louis, MI, USA) were mixed for 20 min. The prepared suspension was coated onto a titanium foil (MF-Ti-Foil-700L-105, MTI Corporation) current collector of 1 cm × 2 cm. Then, the prepared electrodes were dried at 130 °C for 12 h.

#### 2.3.2. Electrochemical Measurements

The capacitive characteristics of CCP, ACP-700, and ACP-800 samples were tested using the Potentiostat P-40X with an FRA-24M electrochemical impedance measurement. The selected electrolyte solution was 6 M KOH in a two-electrode cell. In the symmetric electrochemical cell device, two identical electrodes were placed and separated with a separator. Different scanning rates ranging between 5 to 160 mV/s were applied during the cyclic voltammetry (CV) tests, where the potential window was 0.0–1.0 V. Galvanostatic charge–discharge tests were applied at current densities of 0.1, 0.25, 0.5, 1.0, and 2.0 A/g with a potential window of 0.0–1.0 V [38].

Equation (1), used to derive the specific capacitance (Cs), was applied to the cyclic voltammetry data obtained from the two-electrode cell in order to calculate its value.
(1)$$C_{s}=\frac{A}{m\times k\times (V_{2}-V_{1})}$$
where A is the integral of the CV curve (C); m is the mass of the active substance of one electrode (g); k is the scan rate (mV/s); V_2_ − V_1_ is the potential window (V).

The specific capacity was determined from the galvanostatic charge–discharge (GCD) profiles of the two-electrode cell, employing a designated equation for computation:(2)$$C_{s}=\frac{2I\times t}{m\times \left(V_{2}-V_{1}\right)}$$
where I is the current density (A/g); t is the discharge time (s); m is the mass of the active substance of one electrode (g); V_2_ − V_1_ is the potential window (V).

Impedance measurements were conducted across a frequency range of 300 kHz to 10 MHz, applying an alternating voltage with a 10 mV voltage amplitude to the electrode.

## 3. Results and Discussion

### 3.1. Activated Carbon Surface Characterization

Surface properties are important for materials for double-layer capacitors. The microstructural morphology of the CCP, ACP-700, and ACP-800 was studied using SEM (Figure 2). A beehive-like skeleton structure consisting of three-dimensional macroporous carbon walls was observed on both carbons. The carbonized material (Figure 2a) possessed pores ranging from 2 µm to 33 µm. A decrease in surface pore size was more pronounced in the activated carbons obtained by KOH activation (Figure 2b,c). The KOH-treated carbons showed a hierarchical structure with more ridges and a uniform connected pore network (Figure 2b,c).

The electrochemical performance greatly depends on the activated carbon’s surface area and pore structure. Nitrogen adsorption/desorption isotherms, pore size distributions, and the carbons’ parameters are presented in Figure 3 and Table 1. The carbon after carbonization (CCP) had a surface area of 20.34 m^2^/g. ACP-800 obtained by KOH activation had a surface area of 2333 m^2^/g with a major microporous structure. The surface area of ACP-800 was more than 100-fold greater than that of the CCP. In the presence of KOH, the decomposition reaction occurred in two processes [39]. The metallic potassium intercalates into the carbon matrix and removes the carbon atoms in crystallites [40]. As the gas (CO and CO_2_) escapes, many pores form, contributing to its higher surface area than the porous carbon without KOH activation. The pyrolysis of cucumber peels by treatment with an activator in KOH improved the specific surface area and porous structure dramatically.

(1)The reaction of carbon with potassium hydroxide:6KOH + 2C → 2K + 3H_2_ + 2K_2_CO_3_(2)The decomposition reaction of potassium carbonate and the reaction of carbon with CO_2_, K_2_CO_3_, and K_2_O:K_2_CO_3_→ K_2_O + CO_2_CO_2_ + C → 2COK_2_CO_3_ + 2C → 2K + 3COC + K_2_O → 2K + CO

With the method of KOH activation, the carbon derived at high temperatures resulted in a more developed porous structure. The total pore volume and mesoporous pore volume increased with the increase in activation temperature. The mesoporous volume of ACP-800 was almost double that of ACP-700. A hierarchical structure was more pronounced in ACP-800 than in ACP-700.

The elemental analysis carried out by EDAX is summarized in Table 2. The carbon content in the ACP-700 and ACP-800 samples was 88 and 92%, respectively, while the carbon content of the sample carbonized at a temperature of 600 °C was only about 68%. High carbon content will increase the conductivity of the material, which will help the material’s overall performance as a supercapacitor. The oxygen content in the CCP was about three-fold greater than that of ACP-700 and two-fold greater than that of APC-800. Additionally, most inorganic impurities (such as Mg, Si, K, P, and S) in the ACPs were present in lower amounts than those in the CCP. At an activation temperature of 800 °C, the inorganic matter was negligible. Additionally, it is essential to note that the yield of ACP-700 and ACP-800 was 16.4% and 13.3%, respectively (Table 2).

The thermal gravimetric curves indicate that the ACPs were more thermal stable than the CCP (Figure 4). For the CCP, the peak at 100 °C is ascribed to the release of water. Groups such as quinone, phenol, and lactone that are attached to the carbon matrix can be released from 200 to 800 °C, resulting in their corresponding dTG peaks [41]. The intense peak at 200 °C in ACP-800 indicates that its surface was more oxidized than that of ACP-700. These results are consistent with the elemental composition from EDAX, which showed a higher oxygen percentage for ACP-800. The peak at 680 °C in the ACP-700 sample is attributed to the partial gasification of the least thermally stable fragments of the carbon structure, where the remaining oxygen-containing groups continue to decompose. The carbon skeleton could also react with the released oxygen-containing species at this temperature. However, ACP-800 showed a similar partial gasification around 720 °C. There was a total mass loss of 9.9% and 11.6% for ACP-700 and ACP-800, respectively. From the other thermal stability study [36], the carbons from cucumber peel activated at 800 °C and 900 °C showed excessive total mass losses of 65% and 76%, respectively. The ACP samples demonstrated high thermal stability and high resistance to mass loss.

FT-IR analysis was performed on the carbons, and the spectra are given in Figure 5. More functional groups were observed for the CCP sample than for those activated at 700 °C and 800 °C. The broadband between 3550–3200 cm^−1^ is ascribed to O-H stretching vibrations arising from the moisture content of the material and was observed for all samples [42]. The weak band between 2950–2850 cm^−1^ is assigned to C-H stretching vibrations due to the presence of alkanes and alkenes in the structure of CCP [42]. A strong band around 1620 cm^−1^ was observed for the CCP sample and is assigned to the C=C stretching of sp^2^-hybridized carbon functional groups comprising unsaturated ketones [43]. The medium band between 1440 and 1395 cm^−1^ for the O-H bending vibrations shows the presence of carboxylic acids. C-O stretching vibrations were observed between 1310–1250 cm^−1^ [44]. The overlapping bands in the fingerprint region, at around 850–750 cm^−1^, are ascribed to C-H out-of-plane bending in the aromatic rings. The intensity of all these bands decreased after the activation process, which indicates the decomposition of surface functional groups. After activation at 700 °C, the peaks indicate the presence of O-H (3430 cm^−1^), C=C/O (1610 cm^−1^), and C-O (1400 cm^−1^) on the ACP-700 carbon [45]. A very weak peak for C=C/O (1610 cm^−1^) was also detected for ACP-800. As a result, the number of functional groups decreased with an increase in heat treatment temperature, indicating the removal of functional groups on the surface.

The obtained carbons based on cucumber peel underwent Raman spectra analysis, with the results depicted in Figure 6.

According to Figure 6, the degree of graphitization is determined by calculating the ratio of the G peak area to the total spectrum area between 700 cm^−1^ and 2000 cm^−1^. For instance, the graphitization level is approximately 20% for cucumber peel carbonized at 600 °C, around 22% for cucumber peel carbonized at 600 °C and subsequently activated at 700 °C (600/700), and about 74% for cucumber peel carbonized at 600 °C and activated at 800 °C (600/800).

The degree of graphitization is directly correlated with the width of the G peak in Raman spectroscopy (Table 3). As the activation temperature rises, the degree of graphitization calculated from the spectra also increases, peaking at approximately 74%. These carbonaceous materials, characterized by partial graphitization, demonstrate high electronic conductivities, making them well-suited for electrode applications.

Generally, a narrower G peak indicates a higher degree of graphitization, reflecting a more ordered and crystalline graphite structure. Conversely, a broader G peak suggests lower graphitization, often associated with a greater presence of defects or disordered carbon structures, such as amorphous carbon or disordered graphite. Therefore, the width of the G peak serves as a valuable indicator of the extent of graphitization in carbon materials [46,47].

### 3.2. Electrochemical Characterization

The cyclic voltammetry (CV) of the ACP-700 and ACP-800 electrodes were tested in the potential window of 0.0–1.0 V in a two-electrode cell. The CV curves of both samples (Figure 7 and Figure 8) show a rectangular electric double-layer capacitor (EDLC) shape over a scan rate range of 5–160 mV/s. The rectangular shape of the CV plots at the high scan rate of 160 mV/s indicates fast ion transfer. However, the deviation from the rectangular shape of CVs can be also observed at a high scan rate, indicating a more significant ohmic resistance in the pores [48]. The values of the specific capacitance (Cs) at different scan rates of the material are summarized in Table 1. The ACP-700 sample demonstrated a specific capacitance of 252 F/g at a scan rate of 5 mV/s, while ACP-800 demonstrated a higher specific capacitance of 300 F/g. It is worth noting that there was a decrease in the Cs value with an increase in scan rate. This is due to the lower amount of electrolyte ions exposed to the active areas of the electrode when the scan rate is high [49].

The galvanostatic charge–discharge was measured at various current densities. Figure 9a,b shows typical charge–discharge curves for capacitance behavior. A highly symmetric isosceles triangle at various current densities suggested good electrochemical reversibility and high Coulombic efficiency for both samples. The trend of the gravimetric capacitance of the ACP-700 and ACP-800 electrodes obtained from the GCD measurements is plotted in Figure 9c, and the specific values are shown in Table 2. At 0.1 A/g, ACP-700 had a specific capacity of 237 F/g, while that of ACP-800 was 331 F/g. In Table 4, ACP-800 is compared to other peel-biomass-derived carbons, and it demonstrated excellent specific capacitance with the benefit of a high surface area.

With an increase in the current density to 2 A/g, the gravimetric capacity of ACP-800 remained at 279 F/g, demonstrating a retention of capacity of 84.3%. For ACP-700, the capacitive retention was 93.7%. Both carbons’ high-capacity retention can be linked to their high surface area and hierarchical porous structure, which was beneficial for the electrolyte’s penetration and the electrolyte ions’ diffusion in the electrode at a high current density. Figure 9d presents the cyclic stability of ACP-800 for 2000 charge-discharge cycles at a current density of 1 A/g. The cell with activated carbon electrodes demonstrates stable operation over 2000 cycles in the 0–0.9 V potential range without significant fluctuations.

Figure 10 represents the relationship between the gravimetric capacitance and the square root of the discharge time.

From the Trasatti analysis, gravimetric capacitance (C_g_) can be expressed by the addition of the bilayer capacitance (C_b_) and pseudocapacitance (C_p_) of carbons based on the equation C_g_ = *K*_1_ + *K*_2_*t*^1/2^, in which the rate control *K*_1_ corresponds to the bilayer capacitance (C_b_) and the diffusion control part *K*_2_*t*^1/2^ corresponds to the pseudocapacitance (C_p_) [50]. Figure 10 demonstrates the relationship between the gravimetric capacitance and the square root of the discharge time, with the intercept of the diagonal line on the *y*-axis representing the bilayer capacitance. It was found that the pseudocapacitance contribution of both ACP-700 and ACP-800 was low.

Figure 11a shows a curve of the dependence of Coulombic efficiency on the current density of electrodes made from activated cucumber peel in an aqueous electrolyte with 6 M KOH. The Coulombic efficiency at different current densities was calculated from the GCD curves η = T_d_/Tc × 100 (T_d_ and T_c_ represent the discharge and charge times, respectively). The average Coulombic efficiency value for the ACP-700 electrode was determined as 83.9%, and 89.5% for the ACP-800 electrode at current densities from 0.1 A/g to 2 A/g, while the maximum values of Coulombic efficiency for ACP-700 and ACP-800 were 91.7% at 1 A/g and 94.8 A/g at 2 A/g, respectively.

To study the fundamental behavior of the carbon in EDLCs, electrochemical impedance spectroscopy was performed by using an open-circuit potential over a frequency range of 100 kHz to 10 MHz. The Nyquist plot is shown in Figure 11b.

At very high frequencies, the actual resistance (*x*-axis) comes from the bulk electrolyte. The values of both carbons were around 0.01 Ω, which indicated low resistance of the electrolyte. At a medium–high frequency, the diameter of the semicircle represents charge transfer resistance (R_ct_), which can be correlated with electrical conductivity [51] or the porous structure of carbon [52]. The charge transfer resistance (R_ct_) of the system was approximately 0.07 Ω for ACP-700 and 0.025 Ω for ACP-800. The low resistance of charge transfer of ACP-800 can be due to a more developed porous structure and high electrical conductivity due to the high activation temperature; thus, the rate of electron diffusion increased. The ion diffusion rate is affected by the porous structure of carbon, which is reflected by the slope in the Nyquist plot in the low-frequency region. The larger slope angle of the straight line in the low-frequency region indicates a faster ion diffusion in the electrolyte to the electrode interface for ACP-800 [53,54,55].

## 4. Conclusions

Micro-mesoporous carbon was successfully synthesized by an economical and simple method using cucumber peel biowaste material. The electrochemical properties of the carbons showed potential as effective energy storage materials. In the galvanostatic charge-discharge measurement, the carbon activated at 800 °C (ACP-800) showed a high specific capacitance value of 331 F/g at the current density of 0.1 A/g. The carbon activated at 700 °C (ACP-700) showed 237 F/g in the same conditions. The specific discharge capacitance of the former carbon was 286 F/g at a current density of 1A/g, exhibiting 100% capacity retention after 2000 cycles. SEM images showed that the surface of the carbons formed ridges and pore openings. The specific surface area of the sample activated at 800 °C was 2333 m^2^ g^−1^, determined using the Brunauer–Emmett–Teller theory, which is higher than that of carbon activated at 700 °C (1878 m^2^ g^−1^). The activation process increased the carbons’ surface area dramatically and decreased the number of oxygen species and metal impurities. The high surface area and meso-micropore volume ratio of the sample activated at 800 °C led to its low charge transfer resistance and faster ion transport than that of the sample activated at 700 °C. This activated carbon was produced through KOH chemical activation, which creates a unique three-dimensional porous structure consisting of both micro and mesopores. This structure is believed to be a result of the specific chemical activation process used. These interconnected nanoporous networks provide efficient pathways for electrolyte ions, suggesting that this material could facilitate rapid ionic transport within its hierarchical texture.

## Figures and Tables

**Figure 1 nanomaterials-14-00686-f001:**
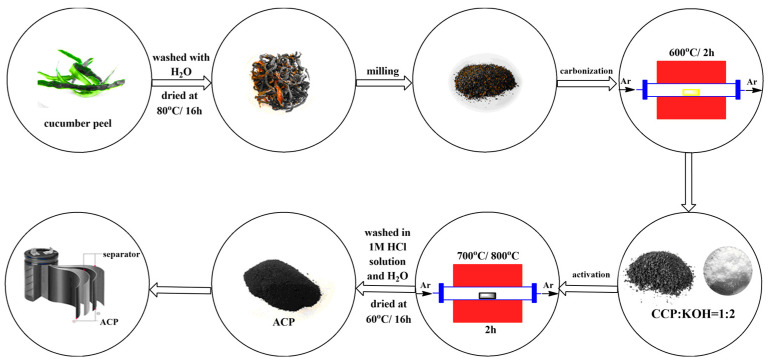
Scheme of the preparation of the carbon materials.

**Figure 2 nanomaterials-14-00686-f002:**
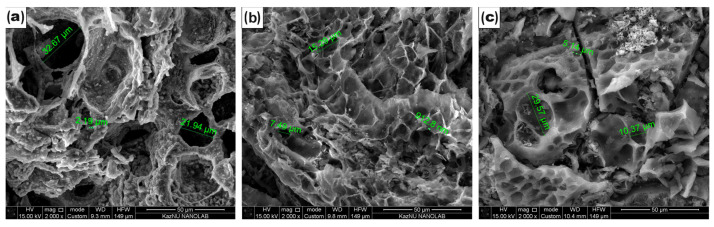
SEM images of (**a**) CCP, (**b**) ACP-700, and (**c**) ACP-800.

**Figure 3 nanomaterials-14-00686-f003:**
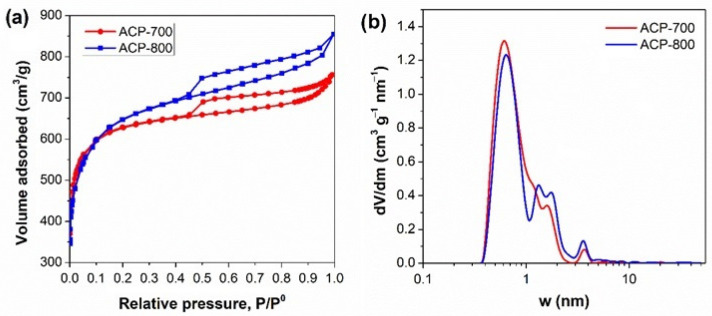
(**a**) N_2_ adsorption isotherm and (**b**) pore size distribution of the ACs.

**Figure 4 nanomaterials-14-00686-f004:**
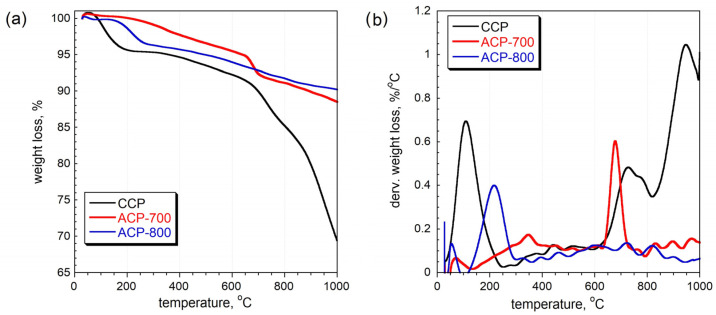
(**a**) Thermal gravimetry (TG) curves and (**b**) differential thermal gravimetry (DTG) curves measured in nitrogen [36].

**Figure 5 nanomaterials-14-00686-f005:**
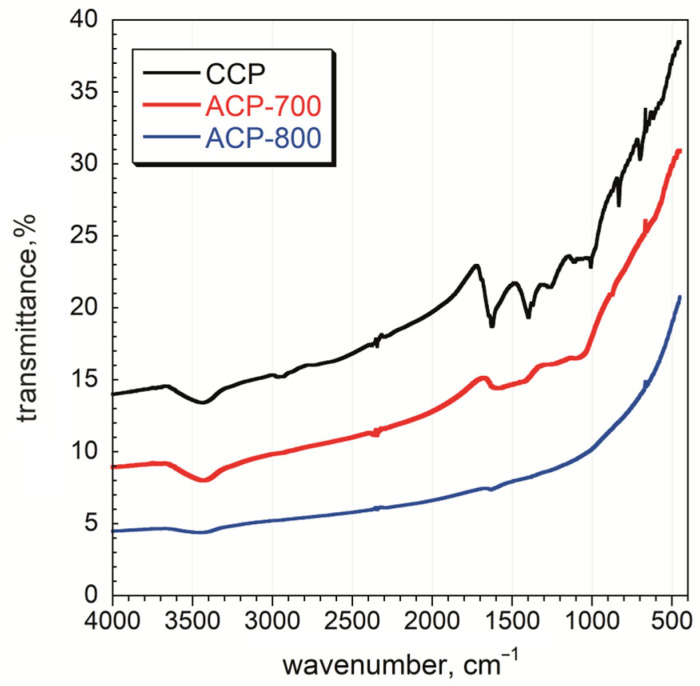
FT-IR spectra of char and activated carbon samples.

**Figure 6 nanomaterials-14-00686-f006:**
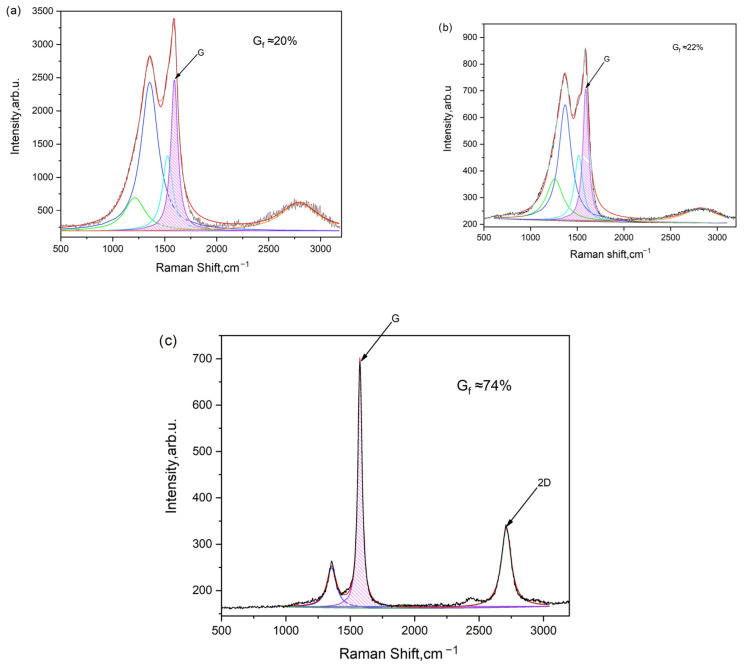
Raman spectra of carbonized and activated carbon samples: (**a**) CCP 600 °C, (**b**) ACP 600/700, and (**c**) ACP 600/800.

**Figure 7 nanomaterials-14-00686-f007:**
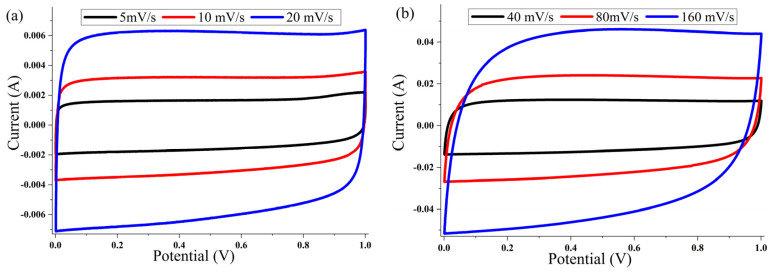
Cyclic voltammetry curves for ACP-700 electrodes with 6 M KOH electrolyte at different scanning speeds: (**a**) 5, 10, and 20 mV/s; (**b**) 40, 80, and 160 mV/s.

**Figure 8 nanomaterials-14-00686-f008:**
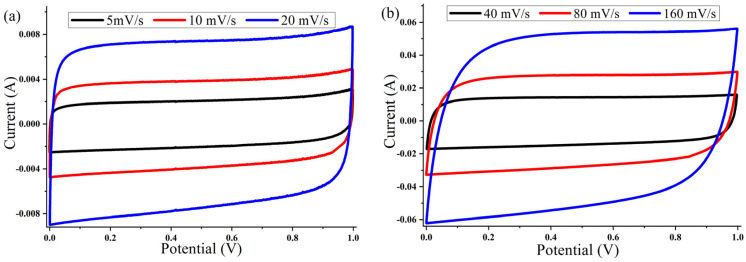
Cyclic voltammetry curves for ACP-800 electrodes with 6 M KOH electrolyte at different scanning speeds: (**a**) 5, 10, and 20 mV/s; (**b**) 40, 80, and 160 mV/s.

**Figure 9 nanomaterials-14-00686-f009:**
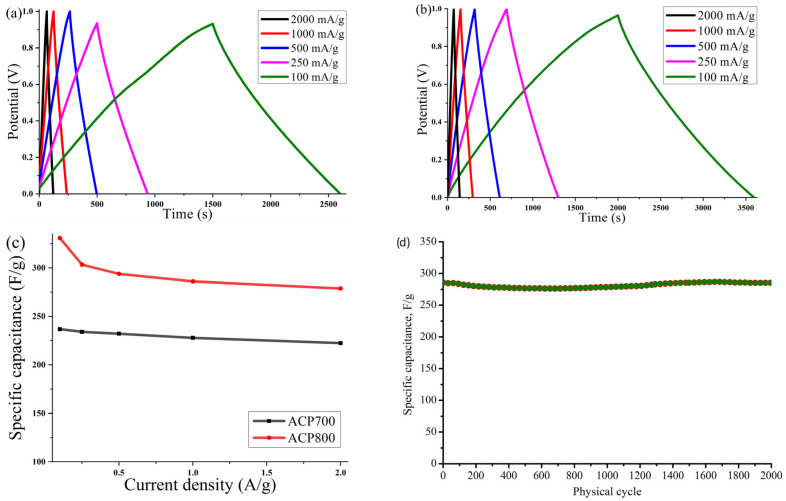
Curves of galvanostatic charge–discharge at different current densities (100, 250, 500, 1000, and 2000 mA/g) for electrodes based on (**a**) ACP-700 and (**b**) ACP-800; (**c**) value of the specific capacitance of the electrodes measured at various current densities; (**d**) cyclic stability of ACP-800 for 2000 cycles at the current density of 1 A/g.

**Figure 10 nanomaterials-14-00686-f010:**
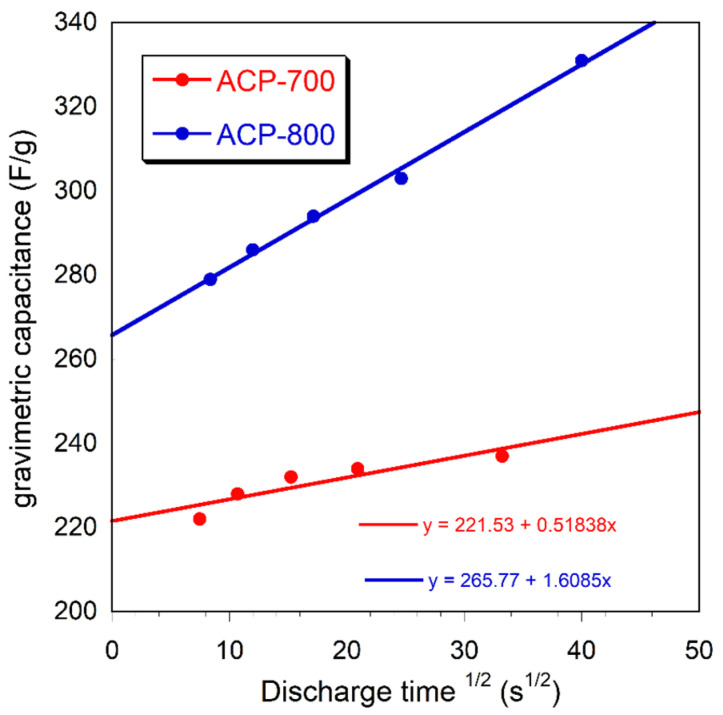
The relationship between the gravimetric capacitance and the square root of the discharge time.

**Figure 11 nanomaterials-14-00686-f011:**
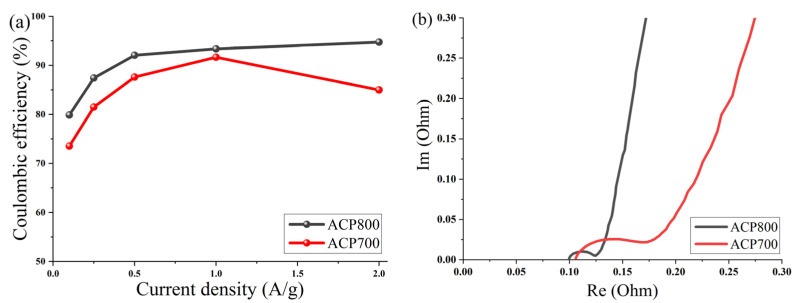
(**a**) Coulombic efficiency as a function of current density for activated cucumber peel electrodes in 6 M KOH and (**b**) Nyquist plot for activated cucumber peel electrodes.

**Table 1 nanomaterials-14-00686-t001:** The parameters of the pore structure of ACPs. The pore volumes are calculated based on the NLDFT method [37].

Sample	S_BET_ (m^2^/g)	V_t_ (cm^3^/g)	V_meso_ (cm^3^/g)	V_<0.7 nm_ (cm^3^/g)	V_<1 nm_ (cm^3^/g)	V_mic_ (cm^3^/g)	V_meso_/V_t_
ACP-700	1878	1.077	0.196	0.242	0.542	0.881	0.18
ACP-800	2333	1.225	0.376	0.237	0.488	0.849	0.31

**Table 2 nanomaterials-14-00686-t002:** Elemental composition and yield of CCP and ACP (atomic concentration, at.%).

Sample	Yield (%)	C	O	Na	Mg	Al	Si	K	Ca	Cl	P	S
CCP	30.9	67.64	20.02	0.23	0.39	0.07	0.42	9.60	-	0.47	0.98	0.17
ACP-700	16.4	91.69	7.04	0.33	0.04	0.07	0.12	0.02	0.16	0.55	-	-
ACP-800	13.3	88.21	11.79	-	-	-	-	-	-	-	-	-

**Table 3 nanomaterials-14-00686-t003:** Correlation of graphitization degree with FWHM (G) and FWHM (D) peaks.

Temperature (°C)	Width of G Peak	Width of D Peak	Degree of Graphitization
600 carbon	8,093,769	20,151,057	20%
600/700	7,347,576	16,164,958	22%
600/800	3,906,941	8,845,874	74%

**Table 4 nanomaterials-14-00686-t004:** Comparison of ACP-800 to other peel-biomass-derived activated carbons.

Carbon Source	S_BET,_ m^2^/g	Electrolyte	Specific Capacitance, F/g	Current Density, A/g	Ref.
Mangosteen peel	1039	3 M KOH	182	0.5	[6]
Cassava peel	1352	0.5 M H_2_SO_4_	264	1	[7]
Cassava peel	-	1 M H_2_SO_4_	183	1	[8]
Watermelon Peel	-	6 M KOH	135	1	[9]
Onion peel	-	1 M HCl	127	0.75	[10]
Orange peel	2160	6 M KOH	226	0.5	[12]
Orange peel	1514	6 M KOH	255	0.5	[13]
Pomelo peel	1582	6 M KOH	180	0.5	[14]
Garlic peel	787	1 M H_2_SO_4_	204	1	[15]
Banana peel	1362	1 M NaNO_3_	165	0.5	[16]
Banana peel	2452	1 M Na_2_SO_4_	220	0.5	[17]
Orange peel	912	1 M Na_2_SO_4_	376	1	[18]
Cucumber peel	2333	6 M KOH	286	1	[this work]

## Data Availability

Data are contained within the article.

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
