# Peer review of "Activated Carbon Derived from Cucumber Peel for Use as a Supercapacitor Electrode Material"

_nanomaterials, 2024, doi:10.3390/nano14080686_

Round 1

Reviewer 1 Report

Comments and Suggestions for Authors

Authors reported Activated Carbon Derived from Cucumber Peel for Use as a Supercapacitor Electrode Material. As a result, I believe that this review work can be published in this journal when the following issues are addressed

1.      In the introduction, some related articles reporting the same methods and ideas should be compared to improve the literature review.

2.      The quality of the Figures is too low. Authors are suggested to improve the quality of the Figures and insets.

3.      Coulombic efficiency is one of the important factors in supercapacitors. Authors should provide and discuss it.

4.      Authors are suggested to add the equations used for calculations.

5.      Some grammatical mistakes were found. The authors need to improve the English of the article.

6.      Pay attention to the format of the text. The format of many references is inconsistent.

Comments on the Quality of English Language

Some grammatical mistakes were found. The authors need to improve the English of the article

Author Response

AUTHORS RESPONSES TO THE REVIEWERS’ COMMENTS

The authors would like to thank the esteemed reviewers for their time spent carefully reviewing the manuscript, and for their valuable comments and feedback.

The comments of the reviewers regarding the submitted manuscript are appreciated. We have revised the manuscript by taking into consideration all reviewers’ comments, and we believe that they have greatly enhanced the quality of our manuscript. Each comment has been addressed point-by-point and the response to reviewer queries is given below.

 REVIEWER I:

 Comment 1: In the introduction, some related articles reporting the same methods and ideas should be compared to improve the literature review.

Response 1: Thank you for pointing this out. In the introduction, related articles reporting the same methods and ideas were compared by giving the characteristics of the carbonaceous materials and the capacitive performances.

Comment 2: The quality of the Figures is too low. Authors are suggested to improve the quality of the Figures and insets.

Response 2:  The resolution of the figures was increased and all figures were revised to improve their quality.

 Comment 3. Coulombic efficiency is one of the important factors in supercapacitors. Authors should provide and discuss it.

Response 3: Thank you for pointing this out. We have added a new graph of Coulombic efficiency vs. current density as Fig.11, a. on the Page.17. Also, this figure was explained in detail under the section 3.2.

 Comment 4:  Authors are suggested to add the equations used for calculations.

Response 4: The equations used for the calculations were included under section 2.3.2, on page 7 as Eq. (1) and (2). The abbreviations in the equations were explained in detail.

Comment 5:   Some grammatical mistakes were found. The authors need to improve the English of the article.

Response 5: The English of the paper was improved, and the paper was controlled by a native speaker.

 Comment 6:   Pay attention to the format of the text. The format of many references is inconsistent.

Response 6:   The references are automatically created. If there are still any problems, they will be checked and corrected before publication.

Reviewer 2 Report

Comments and Suggestions for Authors

Reviewer #: Title: Activated Carbon Derived from Cucumber Peel for Use as a Supercapacitor Electrode Material

This paper is about the preparation of supercapacitor electrode material. Micro-mesoporous activated carbons (AC) are prepared by extracting activated carbon from cucumber peel to improve capacitor properties. I would like to recommend the minor revision, please check the following comments: 

1)     Fig.8 and Fig.9 appear twice in Chapter 3.2, please modify them.

2)     In the second set of Fig. 8c, compared with ACP-700, ACP-800 has A more obvious change in the specific capacitance measured at the current density of 0.25A/g. Is it due to its more developed porous structure?

3)     For a better understanding of the energy storage performance of supercapacitors, whether the relationship between the specific capacitance and the square root of the discharge time has been explored, please carry out further analysis.

4)Has the charge/discharge efficiency at different current densities been explored?

Author Response

The authors would like to thank the esteemed reviewers for their time spent carefully reviewing the manuscript, and for their valuable comments and feedback. The comments of the reviewers regarding the submitted manuscript are appreciated. We have revised the manuscript by taking into consideration all reviewers’ comments, and we believe that they have greatly enhanced the quality of our manuscript. 

 REVIEWER II:

This paper is about the preparation of supercapacitor electrode material. Micro-mesoporous activated carbons (AC) are prepared by extracting activated carbon from cucumber peel to improve capacitor properties. I would like to recommend the minor revision, please check the following comments:

Comment 1: Fig.8 and Fig.9 appear twice in Chapter 3.2, please modify them.

 Response 1: Thank you for pointing this out. We have modified Fig.8 and Fig.9 as requested.   Moreover, we have revised numbering of the figures in the text.

Comment 2:  In the second set of Fig. 8c, compared with ACP-700, ACP-800 has A more obvious change in the specific capacitance measured at the current density of 0.25A/g. Is it due to its more developed porous structure?

Response 2: ACP-800 demonstrated a consistently higher specific capacitance at all measured current densities when compared to ACP-700. As the reviewer mentioned this difference in specific capacitance is primarily due to the porous structure of the samples, as illustrated in Figure 3 and Table 1.

Comment 3:      For a better understanding of the energy storage performance of supercapacitors, whether the relationship between the specific capacitance and the square root of the discharge time has been explored, please carry out further analysis.

Response 3: Thank you for your suggestion

We have added explanation mentioned below to the manuscript at the Page. 16. Fig.10.:

From Trasatti analysis, gravimetric capacitance (Cg) can be express by the addition of the bilayer capacitance (Cb)and pseudocapacitance (Cp) of carbons based on the equation Cg= K1 + K2t1/2, in which the rate control part K1 corresponding to the bilayer capacitance (Cb) and the diffusion control part K2t1/2 corresponding to the pseudocapacitance (Cp) [55 reference was added]. Fig.10 demonstrated the relationship of the gravimetric capacitance and the square root of the discharge time, with the intercept of the diagonal line on the y-axis representing the bilayer capacitance. It was found that the pseudocapacitance contribution of both ACP-700 and ACP-800 was low.

Comments 4:   Has the charge/discharge efficiency at different current densities been explored?

Response 4: The efficiency at different current densities have been explored and added to the manuscript at the Page 17.

Reviewer 3 Report

Comments and Suggestions for Authors

This manuscript deals with supercapacitors with the electrodes based on micro-mesoporous carbon, which was successfully synthesized by an economical and simple method using biowaste material cucumber peels. The electrochemical properties were derived from electrochemical impedance, cyclic voltammetry and galvanostatic charging/discharging. This manuscript is well written. The Introduction section is clear and concise. The experimental part is well presented. This manuscript can be publishable in Nanomaterials.

Maybe, the authors could comment the Nyquist plots of the impedances in a nore extensive form. The Nyquist plots are similar to those obtained form commercial supercapacitors and obey the properties derived in:

- A.A. Moya, Journal of Power Sources, 397 (2018) 124-133.

Author Response

The authors would like to thank the esteemed reviewers for their time spent carefully reviewing the manuscript, and for their valuable comments and feedback. The comments of the reviewers regarding the submitted manuscript are appreciated. We have revised the manuscript by taking into consideration all reviewers’ comments, and we believe that they have greatly enhanced the quality of our manuscript. 

REVIEWER III:

This manuscript deals with supercapacitors with the electrodes based on micro-mesoporous carbon, which was successfully synthesized by an economical and simple method using biowaste material cucumber peels. The electrochemical properties were derived from electrochemical impedance, cyclic voltammetry and galvanostatic charging/discharging. This manuscript is well written. The Introduction section is clear and concise. The experimental part is well presented. This manuscript can be publishable in Nanomaterials.

Comment  1. Maybe, the authors could comment the Nyquist plots of the impedances in a nore extensive form. The Nyquist plots are similar to those obtained form commercial supercapacitors and obey the properties derived in:

- A.A. Moya, Journal of Power Sources, 397 (2018) 124-133.

Response 1:  Thank you for high assessment of our manuscript. We have cited to the mentioned reference as reviewer noted (A.A. Moya, Journal of Power Sources, 397 (2018) 124-133) at the Page. 17 (number of references is 55). But we hope explanation which we presented in the manuscript enough (at the Page 17).

Reviewer 4 Report

Comments and Suggestions for Authors

In this work, the authors prepared activated carbon with high specific surface area and micro mesopores structure using cucumber skin as raw material, and explored its excellent electrochemical performance as an electrode material for supercapacitors in a dual electrode system.However, there are still some issues to consider.

1. In the first paragraph of the introduction, supercapacitors are a new type of energy storage device that falls between traditional capacitors and rechargeable batteries. It is recommended to condense the relevant statement in the background section and refer to relevant literature, such as:

https://doi.org/10.1016/j.cej.2023.143410

https://doi.org/10.1002/aenm.202202286

2. The prepared activated carbon material only underwent a second heat treatment at 700 ℃ and 800 ℃, with insufficient temperature gradient and at least three sets of experimental conditions. Suggest adding an additional set of 900 ℃ conditions to fully study the effect of temperature on the performance of carbonization products.

3. The labeling of aperture in Figure 2 of the text is not clear enough, and the CCP curve in Figure 3b is not displayed. Please carefully check if there are similar issues.

4. The second Figure 8 in the text should be Figure 10, and the second Figure 9 should be Figure 11. Please verify the relevant descriptions of the figures and tables in the article.

5. Figure 6 can be more aesthetically pleasing, with complete borders. Other similar figures should also pay attention to this issue.

6. The yield value format of ACP-800 in Table 2 is incorrect and not aligned in the center. Please verify if other table formats are standardized.

7. The second paragraph on page 11 only briefly describes the graphitization level of each sample in Figure 6. Is there any other characterization method to further prove the defect and degree of graphitization. Please add some possible reasons or conclusions.

Author Response

The authors would like to thank the esteemed reviewers for their time spent carefully reviewing the manuscript, and for their valuable comments and feedback. The comments of the reviewers regarding the submitted manuscript are appreciated. We have revised the manuscript by taking into consideration all reviewers’ comments, and we believe that they have greatly enhanced the quality of our manuscript. 

REVIEWER IV:

In this work, the authors prepared activated carbon with high specific surface area and micro mesopores structure using cucumber skin as raw material, and explored its excellent electrochemical performance as an electrode material for supercapacitors in a dual electrode system. However, there are still some issues to consider.

Comment 1: In the first paragraph of the introduction, supercapacitors are a new type of energy storage device that falls between traditional capacitors and rechargeable batteries. It is recommended to condense the relevant statement in the background section and refer to relevant literature, such as:

https://doi.org/10.1016/j.cej.2023.143410

https://doi.org/10.1002/aenm.202202286

Response 1:  Thank you for pointing this out. A new sentence was written on this issue. We have also referred to literature as reviewer recommended and the background of this statement was condensed.

Comment 2: The prepared activated carbon material only underwent a second heat treatment at 700 ℃ and 800 ℃, with insufficient temperature gradient and at least three sets of experimental conditions. Suggest adding an additional set of 900 ℃ conditions to fully study the effect of temperature on the performance of carbonization products.

Response 2:  The suggestion of the reviewer is highly appreciated by the authors. It would be advantageous to study 900 oC activation temperature to investigate the effect of final temperature on both activated carbon characteristics and capacitive performance. However, due to time limitations for the revision, it is not possible to conduct new experiments.

The reason for not adapting 900 oC activation is that the TGA curves showed a similar thermal stability behavior for 700 and 800 oC ACs, indicating the activation process is nearly completed at these temperatures. On the other hand, it is known that the higher the activation temperature, the better the surface porosity/carbon content and the lower the product yields. So, authors performed the experiments at 700 and 800 oC to achieve moderately high C content and porosity with less mass loss.

Comment 3: The labeling of aperture in Figure 2 of the text is not clear enough, and the CCP curve in Figure 3b is not displayed. Please carefully check if there are similar issues.

Response 3: Thank you for pointing this out. We have revised Figures 2-3 and the typing errors were corrected in the figures.

Comment 4:  The second Figure 8 in the text should be Figure 10, and the second Figure 9 should be Figure 11. Please verify the relevant descriptions of the figures and tables in the article.

Response 4:  Thank you for notes. We have revised numbering of the figures in the text.

Comment 5:  Figure 6 can be more aesthetically pleasing, with complete borders. Other similar figures should also pay attention to this issue.

Response 5:  Figure 6 was redrawn with complete borders.

Comment 6:  The yield value format of ACP-800 in Table 2 is incorrect and not aligned in the center. Please verify if other table formats are standardized.

Response 6:  Thank you for pointing this out. The yield value format of ACP-800 in Table 2 corrected and aligned in the center and other tables also checked.

Comment 7:  The second paragraph on page 11 only briefly describes the graphitization level of each sample in Figure 6. Is there any other characterization method to further prove the defect and degree of graphitization. Please add some possible reasons or conclusions.

Response 7:  To clarify the graphitization degree in a better manner, Raman spectra of the samples were included in the study. A new table was prepared and given as Table 3. The graphitization degree was explained in detail with the help of G and D bands obtained from Raman spectra.

Reviewer 5 Report

Comments and Suggestions for Authors

1.      The scale bar of figure 2 is not clear

2.      Line 226: the reaction is not balanced

3.      Figure 5 : the y axis label should be only Transmittance. Moreover, the curves seem to be not normalized

4.      Figure 9: avoid overlapping curves with legend box

5.      Figure 8c is blurry

6.      XRD analysis should further corroborate the reported results

Comments on the Quality of English Language

Minor editing of English language required

Author Response

The authors would like to thank the esteemed reviewers for their time spent carefully reviewing the manuscript, and for their valuable comments and feedback. The comments of the reviewers regarding the submitted manuscript are appreciated. We have revised the manuscript by taking into consideration all reviewers’ comments, and we believe that they have greatly enhanced the quality of our manuscript. 

REVIEWER V:

Comment 1: The scale bar of figure 2 is not clear

Response 1: The resolution of the figure 2 was increased and now scale bar is more clearer.

Comment 2: Line 226: the reaction is not balanced

Response 2: Thank you! We have corrected this typing error.

  Comment 3: Figure 5: the y axis label should be only Transmittance. Moreover, the curves seem to be not normalized

Response 3: In Figure 5 the y axis label was changed as “Transmittance, (%)”. The automatic baseline correction was carried out the FTIR spectra using the OMNIC software. The baseline was not corrected manually not to miss any peaks.

Comment 4: Figure 9: avoid overlapping curves with legend box.

Response 4: Thank you! We have redrawn figures.

Comment 5: Figure 8c is blurry

Response 5: The resolution of the figures was increased, and all figures were revised to improve their quality. Moreover, we have revised numbering of the figures in the text.

Comment 6: XRD analysis should further corroborate the reported results.

Response 6: Thank you for your suggestion. XRD analysis gives a important information about the crystal structure and graphitization degree of the carbonaceous materials, which can be confirmed also by Raman spectra. We agree with the reviewer that the corroboration of the characterization results would be better if the XRD analysis was included in this article. Since the revision time is limited, although we would like to add this analysis in our study, it is not possible to conduct the experiments till the revision due date. But, this suggestion is very valuable for our next articles.

From Table 2, the majority of elements from the elemental analysis was carbon (> 88%). The XRD of amorphous carbon will have a typical broad peak at 2θ =15-30º, which corresponds to 002 plane.   

Round 2

Reviewer 4 Report

Comments and Suggestions for Authors

The authors have addressed all of my concerns, and I am glad to accept it in its present form.